Relationship between circadian genes and memory impairment caused by sleep deprivation

Ke Peng 1 2
Zheng Chengjie 2
Liu Feng 2
Wu LinJie 3
Tang Yijie 2
Wu Yanqin 2
Lv Dongdong 2
Chen Huangli 2
Qian Lin 2
Wu Xiaodan wxiaodan@sina.com 2
Zeng Kai fymzk6822@163.com 1
1 Department of Anesthesiology, Anesthesiology Research Institute, the First Affiliated Hospital of Fujian Medical University , Fuzhou , Fujian , China
2 Department of Anesthesiology, Shengli Clinical Medical College, Fujian Medical University , Fuzhou , Fujian , China
3 Institute of Pharmaceutics, College of Pharmaceutical Sciences, Zhejiang University , Hangzhou , Zhejiang , China
Uversky Vladimir
Electronic publication date: 2022 Mar 21
Publication date: 2022
Volume: 10
Electronic Location ID: e13165
Received 2021 Dec 27; Accepted 2022 Mar 4
Copyright: ©2022 Ke et al.
Copyright year: 2022
Copyright holder: Ke et al.
License: This is an open access article distributed under the terms of the Creative Commons Attribution License, which permits unrestricted use, distribution, reproduction and adaptation in any medium and for any purpose provided that it is properly attributed. For attribution, the original author(s), title, publication source (PeerJ) and either DOI or URL of the article must be cited.
License URL: https://creativecommons.org/licenses/by/4.0/

Keywords: Memory, Sleep deprivation, Immune environment, Hippocampus, Circadian gene

Funding: The Natural Science Foundation of Fujian Province, China 2019Y9013 2020J011080 This work was supported by the Natural Science Foundation of Fujian Province, China (Grant No. 2019Y9013 and No. 2020J011080). The funders had no role in study design, data collection and analysis, decision to publish, or preparation of the manuscript.

==============================
Background

Sleep deprivation (SD)-induced cognitive impairment is highly prevalent worldwide and has attracted widespread attention. The temporal and spatial oscillations of circadian genes are severely disturbed after SD, leading to a progressive loss of their physiological rhythms, which in turn affects memory function. However, there is a lack of research on the role of circadian genes and memory function after SD. Therefore, the present study aims to investigate the relationship between circadian genes and memory function and provide potential therapeutic insights into the mechanism of SD-induced memory impairment.

Methods

Gene expression profiles of GSE33302 and GSE9442 from the Gene Expression Omnibus (GEO) were applied to identify differentially expressed genes (DEGs). Subsequently, both datasets were subjected to Gene Set Enrichment Analysis (GSEA) to determine the overall gene changes in the hippocampus and brain after SD. A Gene Oncology (GO) analysis and Protein-Protein Interaction (PPI) analysis were employed to explore the genes related to circadian rhythm, with their relationship and importance determined through a correlation analysis and a receiver operating characteristic curve (ROC), respectively. The water maze experiments detected behavioral changes related to memory function in SD rats. The expression of circadian genes in several critical organs such as the brain, heart, liver, and lungs and their correlation with memory function was investigated using several microarrays. Finally, changes in the hippocampal immune environment after SD were analyzed using the CIBERSORT in R software.

Results

The quality of the two datasets was very good. After SD, changes were seen primarily in genes related to memory impairment and immune function. Genes related to circadian rhythm were highly correlated with engagement in muscle structure development and circadian rhythm. Seven circadian genes showed their potential therapeutic value in SD. Water maze experiments confirmed that SD exacerbates memory impairment-related behaviors, including prolonged escape latencies and reduced numbers of rats crossing the platform. The expression of circadian genes was verified, while some genes were also significant in the heart, liver, and lungs. All seven circadian genes were also associated with memory markers in SD. The contents of four immune cells in the hippocampal immune environment changed after SD. Seven circadian genes were related to multiple immune cells.

Conclusions

In the present study, we found that SD leads to memory impairment accompanied by changes in circadian rhythm-related genes. Seven circadian genes play crucial roles in memory impairment after SD. Naïve B cells and follicular helper T cells are closely related to SD. These findings provide new insights into the treatment of memory impairment caused by SD.

Introduction

Sleep plays a vital role in the human nervous system (Keene & Duboue, 2018). Adequate sleep time is essential for the clearance of toxic metabolites produced by the body, nerve injury self-repair, and cell stress reduction (Hossain et al., 2021), preventing the occurrence of a variety of diseases. However, with the rapid pace of modern society, healthy sleep patterns are hard to achieve, and SD has become a common phenomenon. Numerous studies have shown that SD has deleterious effects on cognitive function by impairing neurological functions and development (Sabia et al., 2021) contributing to a decrease in attention, working memory, emotional signal transmission speed in the brain, and reaction time (Turner et al., 2007). SD can also lead to neurodegenerative diseases and aging (Zada et al., 2021). In addition, SD also suppresses immune and cardiovascular functions (Irwin, 2015). Short-term SD impairs short-term memory and spatial learning memory (Dai et al., 2021), while long-term SD damages novelty-related object recognition memory and object location memory (Jiao et al., 2021). Memory impairment can make daily living significantly harder and can lead to serious adverse socio-economic consequences. Therefore, it is important to clarify the specific mechanism of memory impairment caused by SD.

Circadian clock genes, a kind of gene capable of 24 h autonomous periods and driven by transcription-translation feedback loops (Ikegami et al., 2019), control daily activities and maintain organ function in many physiological processes (Patke et al., 2017). Circadian clock genes contains the core clock genes (such as Arntl, Clock, Per1, Per2, Per3, Cry1, Cry2, and Cry3) and the clock control genes (such as Rora, Rorb, Rorc, Nr1d1, and Nr1d2). In recent years, studies have shown that circadian rhythm plays an important role in many neurological disorders, including schizophrenia, bipolar disorder, depression, and autism (Johansson et al., 2016). The direct inhibition or activation of circadian clock genes plays a regulating role in diseases. For example, patients with Alzheimer’s disease (AD) lose their physiological rhythm of circadian genes, which aggravates cognitive impairment (Ni et al., 2019). Interestingly, the disorder of circadian genes affected by SD can also exacerbate the neuropathological damage of AD (Niu et al., 2021). Bmal1, found in the brain and muscle, is one of the most classic circadian genes, and participates in the hippocampal-dependent memory process by regulating the phagocytosis of microglia (Wang et al., 2021a). In addition, Dicer1 is one of the genetic factors that influence the expression of Cry1 (Lee et al., 2013) and neuronal plasticity (Tinarelli et al., 2014) after SD. Xbp1 maintains the rhythmicity of gene transcription in the mouse liver (Meng et al., 2020). In short, circadian genes play vital roles in the nervous system and in memory function.

Research on the immune system and its connection to memory has attracted considerable attention (Castellano et al., 2017). Spatial memory function is restored in immunodeficient mice after reconstitution of immune function by splenocytes. Immune cells in the central nervous system (CNS), especially T cells, are essential for maintaining hippocampal plasticity and learning memory (Ziv et al., 2006). Further research shows that CD4 T cells promote memory through the IL-4-driven regulation of synaptic function and long-term potentiation (Herz et al., 2021). In AD, eliminating NK cells enhances neurogenesis and reduces neuroinflammation, improving cognitive function in mice (Zhang et al., 2020). Changes in the immune environment of different parts of the body may also have significance (Kaiser et al., 2014). CD4 and IL-6 in the blood are associated with memory function (Wang et al., 2017). In general, the immune environment is a notable factor in memory.

Circadian genes and the immune environment are two determinants of memory function. However, their changes after SD remain insufficiently explored. Although some studies have shown the relationship between circadian genes and immune cells (Labrecque & Cermakian, 2015), their results are still far from comprehensive. The role of circadian genes and the immune environment in memory impairment caused by SD is still unclear. The present study deepens the understanding of memory impairment in SD through bioinformatics analyses and animal experiments, and provides new insights for further research on circadian genes and immune environments in SD.

Materials & Methods

Acquisition of microarray datasets

The mRNA microarray datasets that met the study requirements were selected by searching for ‘sleep deprivation’ and downloaded from the GEO database (http://www.ncbi.nlm.nih.gov/geo/) (Barrett et al., 2013). The experimental screening conditions required at least eight samples in each group. The eligible GSE33302 (Vecsey et al., 2012) and GSE9442 (Maret et al., 2007) datasets were incorporated, with the former containing nine control samples and eight sleep deprivation samples from the mouse hippocampus and the latter containing three different mouse strains. To ensure the consistency of the results, we opted for mice of the same strain from GSE33302 for the study, including twelve control samples and twelve sleep deprivation samples from the whole brain tissue of the mice. In GSE33302, mice in the sleep deprivation group were subjected to a single 5-hour SD using a gentle method (meaning the researcher made a slight artificial noise or knocked or pushed the animal’s cage, disturbed the animal’s nesting material, or stroked the mice to keep it awake). In GSE9442, mice were subjected to a single 6-hour SD in the same way. In addition, validation datasets (the entire GSE9442 dataset and other related datasets, including GSE98582, GSE42323, GSE92913, and GSE42324) and animal experiments enhanced the reliability of the results. The GSE9442 dataset had a large sample size and included three different species, which extends the generalizability of our results.

Screening of DEGs after SD

To identify changes in mice after SD, GEO2R was applied to analyze the DEGs between the sleep deprivation and control groups. GEO2R is a web tool in the GEO database that compares gene expression profiles through the Limma package (Ritchie et al., 2015), thus efficiently obtaining DEGs between groups. The appropriate datasets were selected, from which samples were classified and nominated, and then analyzed according to the default parameters. Previous studies have shown that small changes in brain tissue genes can have a dramatic physiological impact; therefore, we chose genes with FDR < 0.05 as DEGs.

GSEA enrichment analysis

GSEA enrichment analysis is a gene set-based analysis method that compares genes from the expression profile with predefined functional gene sets. It is used to identify the differences in genes and analyze the functions of the genes that change (Subramanian et al., 2005). Compared with a traditional analysis, a GSEA enrichment analysis can find minor altered genes, especially suitable for this study. The GSEA analysis was performed in WebGestalt, a WEB-based GeneSet Analysis Toolkit (Liao et al., 2019). After selecting ‘Mus Musculus’ and the ‘gene set enrichment analysis (GSEA)’ method, we set ‘functional database’ as ‘pathway’ and ‘KEGG’ to ‘perform GSEA analysis.’ Then, we chose ‘gene symbol’ and submitted all genes and corresponding change values with ‘advanced parameters’ set as the default. After clicking ‘submit,’ the program would return the functional categories of the genes involved.

Circadian rhythm-related genes

The GeneCards database is the most comprehensive gene database, integrating a large amount of literature information and covering multiple databases containing gene information (Stelzer et al., 2016). After logging into GeneCards, we entered ‘circadian rhythm’ to probe circadian rhythm-related genes, and then intersected them with the DEGs of GSE33302 and GSE9442 to obtain the circadian rhythm-related genes.

Functional analysis and PPI analysis of circadian rhythm-related genes

The Gene Ontology (GO), Kyoto Encyclopedia of Genes and Genomes (KEGG) pathways, and PPI network analyses on circadian rhythm-related genes were performed by inputting these genes into Metascape (Zhou et al., 2019) and STRING (Szklarczyk et al., 2021) with ‘mice’ selected as the species. The network was constructed and analyzed to obtain the core modules that participate in disease development, as defined by the Molecular Complex Detection (MCODE) algorithm (Bader & Hogue, 2003).

Correlation of circadian rhythm-related genes with their diagnostic value

In the R software, a Pearson correlation analysis revealed the relationship among the circadian rhythm-related genes. Also, to verify the clinical roles of the most direct circadian genes, we performed an ROC analysis of GSE33302 and GSE9442 (all samples) to investigate their diagnostic and prognostic value in SD.

Experimental animals and models of SD

Male Sprague Dawley rats, eight weeks old, weighing 250–300 g, were used in this study. Animals were purchased from the experimental animal center of Fujian Medical University and housed under standard light (12 h light/dark cycle) and temperature (23 ± 2 °C) conditions in the general laboratory of the experimental animal center of Fujian Medical University. The experimental process was approved in an ethics review by the Committee of experimental animals of the Fujian Medical University and carried out in strict accordance with the animal management regulations and regulations of the Fujian Medical University to maximize the welfare of the animals (FJMU IACUC 2021-0458). Each sleep deprivation and control group contained five randomly assigned rats, as determined by the principle of minimum sample size (n = 5). The rats were acclimated to the environment for two weeks. We have successfully established SD models in the past following Kaushal’s methodology (Kaushal et al., 2012) by using the rat SD instrument. This instrument contains a transverse bar with an adjustable cycle, which rotates and runs along the bottom of the feeding cage from the central point of the cage, forcing the rats to cross the metal bar intermittently. After turning on the SD instrument, the parameters of the deprivation bar were set as follows: the transverse bar rotated ten turns per minute, rotating 1.5 turns counterclockwise after 1.5 turns clockwise, and then rotating 1.5 turns clockwise. At the beginning of the experiment, rats in both groups moved, ate, and drank freely in the rat SD chamber with the SD apparatus. In the sleep deprivation group, the rat SD instrument remained on from light period four to dark period twelve for seven days, which means the rats slept four hours every day. In the control group, the SD chamber remained off all the time. Our previous study demonstrated the successful preparation of animal models. After the experiments, we euthanized the rats in each group to ensure compliance with animal ethics (after intraperitoneal injection of 2% Pentobarbital Sodium at 50 mg/kg, we killed the rats by cervical dislocation). In the sleep deprivation group and in the water maze, we ensured a comfortable living environment and adequate food and water for the experimental animals in order to reduce discomfort and pain.

Morris water maze

The Morris water maze tested the learning memory of the rats. The water maze was done in a circular swimming pool with a diameter of 1.5 m and a depth of 0.45 m. The pool contained four quadrants. A camera was installed above the pool to record and analyze the movement of the rats. During the experiment, the experimenters kept quiet. In the first stage of the Morris water maze, a circular platform with a diameter of 15 cm in the southwest quadrant of the pool was placed two cm lower than the water surface. Each rat was placed in four different quadrants every day and swam freely for 60 s. If the rat mounted the circular platform within 60 s, we allowed it to stay on the circular platform for 15 s and then started it again in the next quadrant. If the rat did not board the circular platform within 60 s, it would be placed on the circular platform for 15 s manually and then started again in the next quadrant. Rats swimming too fast or too slow on the first day were excluded to ensure that the exercise capacity of the rats was consistent in this experiment. We repeated the experiment once a day for four consecutive days. In the second stage, we removed the circular platform in the pool and placed the rats in the northeast quadrant, allowing them to swim freely for 60 s to test their escape latency and platform crossing numbers. The computer recorded the number crossing the platform and the escape time. The data were analyzed independently by an analyst without information about the group.

Verification of circadian genes

To verify the reliability of the results, multiple datasets, including GSE9442 (brain), GSE98582 (human blood) (Uyhelji et al., 2018), GSE42323 (mice heart) (Wormald et al., 2006), GSE92913 (mice liver) (Fan et al., 2021), and GSE42324 (mice lung) (Anafi et al., 2013), were used to verify the expression of circadian genes. First, we performed all samples of GSE9442 to verify the expression of circadian genes. To validate the conclusion that these genes also have potential applications in humans, the human blood sample datasets GSE98582 were applied to observe the changes of these genes after SD. Based on the influence of SD on multiple organs, we explored whether these circadian genes also have possible application value in other organs, including the heart, liver, and lungs.

Circadian genes affect memory function

It was essential to elucidate whether circadian genes affect memory function. Two memory function markers, BDNF and NPAS4, were presented to analyze their correlation with circadian genes. We obtained information about these genes from the expression profiles of GSE9442 (all samples). The circadian genes and memory markers were closely correlated, indicating a relationship exists between circadian genes and memory function.

SD and immune infiltration

CIBERSORT (Newman et al., 2019) is a method used to analyze the possible components of immune cells based on the expression characteristics of each known cell. We calculated the gene expression profile of the mouse hippocampus before and after SD to analyze the immune changes of the mouse hippocampal environment. We applied CIBERSORT to obtain the changes of immune cells in each sample. Finally, the relationship between circadian genes and these immune cells was calculated using a Pearson correlation analysis. GSE34424 (Porter et al., 2012), containing SD and control samples from the hippocampus, was used to verify the results from the correlation analysis of immune cells.

Statistical analysis

A statistical analysis was performed in GraphPad Prism 7.0, GEO2R, and R software. We presented numerical data as the mean ± standard deviation. The statistical significance of the difference was tested using a Student’s T-test. The screening criteria for DEGs, GSEA enrichment analysis results, GO and KEGG functions were FDR < 0.05. A P < 0.05 was considered statistically significant with *, **, *** and **** representing P < 0.05, 0.01, 0.001 and 0.0001, respectively.

Results

Gene expression of microarray datasets

The boxplot results from GEO2R showed that the median, upper quartile, lower quartile, and distribution of all samples in the standardized datasets of GSE33302 and GSE9442 were relatively consistent, indicating the position and dispersion of the data met the quality requirements. Samples from the sleep deprivation group were colored green, and the ones from the control group were colored purple (Figs. 1A; 1D). The gene expression density maps visually described the distribution of gene expression in each sample. The intensity of genes was mainly between 3 and 12, which showed their consistency in their respective datasets and ideal sample data quality (Figs. 1B; 1E). The volcano maps showed DEGs in the two datasets, with up-regulated genes marked in red and the down-regulated genes marked in blue (Figs. 1C; 1F). We displayed these DEGs in ‘Supplemental Information 1.’

Figure 1 Expression of genes between sleep deprivation and control.

(A) Boxplots of GSE33302. Samples from the sleep deprivation group are colored green and the ones from the control group are colored purple. (B) The density map of GSE33302 shows the expression of each sample. (C) The volcano map shows DEGs in GSE33302. The up-regulated genes are colored red and the down-regulated genes are colored blue. (D) Boxplots of GSE9442. (E) The density map of GSE9442. (F) The volcano map of DEGs in GSE9442. DEGs, differentially expressed genes.

Functional enrichment analysis of GSEA on SD

After SD, the gene transcription level of the mouse hippocampus and brain tissue changed. The change of genes in the nervous system was microscopic. However, a minor variation of genes in the nervous system can cause considerable changes in nervous system function. Therefore, we performed a GSEA analysis to explore the effect of SD. In GSE33302, multiple statistically significant items were enriched, including promoting systemic lupus erythematosus and asthma (immune diseases; Fig. 2A). GO and KEGG terms associated with the process of memory formation were enriched on the suppression of long-term potentiation and axon guidance after SD. Also, the inhibition of nicotine addiction, ferroptosis, and oocyte meiosis suggested that SD affects iron death and fertility. Long-term potentiation, the most statistically significant item is shown in detail in Fig. 2B; other terms are presented in ‘Supplemental Information 2.’ In GSE9442, we found a promotion of protein processing in the endoplasmic reticulum, protein export, and transcriptional misregulation in cancer. Unsurprisingly, memory-related items, including nucleotide excision repair and glycosaminoglycan degradation, were significantly suppressed. These results showed that the changes in overall gene-level after SD were mainly related to aggravated memory impairment and immune diseases.

Figure 2 GSEA enrichment on genes between sleep deprivation and control.

(A) GSEA enrichment in GSE33302. Statistically significant functional items are represented by dark blue and dark orange. A dark blue item indicates that SD promotes its progress. (B) Details of long-term potentiation. (C) GSEA enrichment in GSE9442. (D) Details of Glycosaminoglycan degradation.GSEA, gene set anrichment analysis.

Functional analysis of circadian rhythm-related genes after SD

To clarify the function of the genes that changed with SD, 29 differently expressed circadian rhythm-related genes were obtained by overlapping circadian rhythm-related genes (Supplemental Information 3) with DEGs in GSE33302 and GSE9442 (Fig. 3A). A GO and KEGG analysis showed that these genes were mainly involved in muscle structure development, cellular response to extracellular stimulus, selenium metabolism/selenoproteins, and circadian rhythm (Fig. 3B). These functions were closely related to each other (Fig. 3C). The MCODE module in the PPI network, with circadian rhythm genes marked in pink, showed seven circadian genes (Figs. 3D–3E).

Figure 3 Functional enrichment on circadian rhythm-related genes.

(A) The Venn diagram shows 29 differently expressed circadian rhythm-related genes. (B) Bar graph for the GO and KEGG enrichment analysis of these genes, one per cluster, using P < 0.05 to represent statistical significance. (C) Enrichment network clusters of circadian rhythm-related genes show the connection between enriched terms. (D) The PPI network of circadian rhythm-related genes. Genes enriched in circadian rhythm are colored pink. (E) Important module from PPI identified by MCODE analysis.GO, gene oncology; KEGG, Kyoto Encyclopedia of Genes and Genomes; PPI, protein-protein interaction; MCODE, molecular complex detection.

Relationship between circadian rhythm-related genes

A correlation analysis explored the relationship between the 29 circadian rhythm-related genes. The results showed a synergistic or inhibitory relationship between these circadian rhythm-related genes, with seven direct circadian genes showing the strongest relationship (as shown in red boxes in Fig. 4A). To further verify the importance of these seven circadian genes in SD, an ROC analysis observed the role of these genes through the identification of SD injury. Generally, genes with a high AUC will play a decisive role in the pathogenesis of the disease. All seven genes shared high values in GSE33302 (Supplemental Information 4) and GSE9442 (Fig. 4B). ‘Figure 4C’ shows the expression of these seven circadian genes. The above results showed that these seven circadian genes play important roles in SD.

Figure 4 Correlation analysis of circadian rhythm-related genes.

(A) Correlation heat maps show the correlation between circadian rhythm-related genes in GSE33302 and GSE9442. The red indicates a positive correlation and the blue indicates a negative correlation. (B) Value of seven circadian genes in GSE9442. (C) The expression of circadian genes in GSE33302.

Relationship between circadian genes and memory

A GSEA analysis found that SD inhibited memory function. In the water maze experiment, SD significantly reduced the number of rats that crossed the platform (from 1.2 to 3.2) and significantly increased the time it took to escape the platform (from 9.94 s to 39.54 s; Figs. 5A–5B). The trajectory diagram of the rats showed less exploration of the plateau after SD (Fig. 5C), as evidenced by the varied expression of seven circadian genes (Fig. 5D), which was in agreement with GSE33302. Meanwhile, the expression of DICER1 and XBP1 changed after SD in human blood samples of GSE98582 (Fig. 5E). However, due to different sequencing platforms, other genes had no corresponding test results in the datasets. Circadian genes play roles in all organs, which indicates the need for further exploration in various organs. Therefore, we detected the expression of these genes in the heart (GSE42323), liver (GSE92913), and lungs (GSE42324), observing that some genes alternated in various organs (Fig. 5F). Seven circadian genes significantly and consistently correlated with two representative indicators of memory function (BDNF and NPAS4). These results indicate that SD leads to memory impairment and changes in seven circadian genes closely related to memory function.

Figure 5 Cognitive changes and the verification of seven circadian genes.

(A) The number of rats crossing the platform. (B) Time to escape across the platform. (C) Rat trajectory maps show a significantly decreased exploration of the plateau after SD. The expression of circadian genes in the (D) brain, (E) human blood, (F) heart, liver, and lungs between the control and SD groups. (G) Seven circadian genes are significantly associated with BDNF and NPAS4.

SD altered the hippocampal immune infiltration

Previous studies have reported immunosuppression after SD. However, the changes in the hippocampus after SD are still implicit. M2 macrophages, naïve B cells, follicular helper T cells, plasma cells, resting memory CD4 T cells, activated NK cells, CD8 T cells, monocytes, activated dendritic cells, and resting NK cells were all validated as the most prominent cells in the hippocampal immune environment (Fig. 6A). The contents of four immune cells, including naïve B cells, follicular helper T cells, activated NK cells, activated dendritic cells, changed during SD (Figs. 6B–6D), further demonstrating that circadian genes were related to the changes in naïve B cells, follicular helper T cells, activated NK cells, M2 macrophages, monocytes, and activated dendritic cells after SD (Fig. 6E). In addition, naïve B cells were verified to be upregulated after SD in GSE34424 (Supplemental Information 7). These results showed that circadian genes also affected the immunity of the CNS.

Figure 6 Changes in immune infiltration after sleep deprivation.

(A) The proportion of various immune cells in the hippocampus. (B) The heat map shows the expression of immune cells in each sample. (C) The content of different immune cells. (D) Changes of hippocampal immune cells after sleep deprivation show four types of altered cells. (E) Circadian genes are associated with immune cells.

Discussion

As a widespread chronic disease, SD severely damages learning, memory, and immune system function (Xu et al., 2021). The damage caused by sleep deprivation persists even after entering a healthy sleep regime. Although the spatial memory impairment improved 18 h after a healthy sleep regime (Wang et al., 2009a), hippocampal function in the CA1 region remained affected for more than 24 h (Kim, Mahmoud & Grover, 2005). Memory consolidation was still inhibited by sleep deprivation 48 h after normal sleep, showing the long-term effects of sleep deprivation (Li et al., 2009). Inflammatory markers, including IL-17, remained high even after seven days of normal sleep, suggesting that sleep deprivation causes an inflammatory state over a prolonged period (Yehuda et al., 2009). Further study showed that SD causes long-term damage by affecting cortical synapses (Simor et al., 2017). Disturbances in the body’s internal rhythms may be a vital cause of long-term injury. Therefore, it is necessary to study circadian genes. Recently, SD studies have made a series of achievements, with discoveries in memory function and circadian genes (Niu et al., 2021). One study found that changes in the hippocampal region after SD are critical for memory (Gaine et al., 2021). However, there is a lack of research linking the alterations of circadian genes and immune function after SD to memory function. In this study, GSE33302 and GSE9442 were applied for analysis to obtain more reproducible results, both of which showed a high correlation between their changes and memory functions. Differentially expressed circadian genes were obtained and used to analyze their relationship with memory function. Finally, the changes in the immune environment in the hippocampus after SD showed a high correlation with circadian genes.

The SD models of both datasets showed inhibited memory function. Long-term potentiation and axon guidance are functional terms essential for maintaining normal memory function (Frye et al., 2021), both of which were suppressed after SD in GSE33302. In GSE9442, nucleotide excision repair dysfunction inhibited histone recovery, resulting in epigenetic instability and memory impairment (Polo, Roche & Almouzni, 2006). The accumulation of glycosaminoglycan leads to CNS inflammation and increases the permeability of the blood–brain barrier (Wang et al., 2009b). These results highlighted the effect of SD on memory and verified the consistency and reliability of the datasets. SD increases the risk of autoimmune diseases and aggravates their symptoms (Garbarino et al., 2021), but the mechanism behind this relationship still lacks sufficient research. Our GSEA results convincingly concluded that SD affected autoimmune diseases, which were characterized by changes in the immune environment in the hippocampus, providing new ideas for the treatment of immune disorders. Notably, among these GSEA enriched entries, previous studies validated the role of SD on ferroptosis (Wang et al., 2021b), oocyte meiosis (Yong et al., 2021), and transcriptional misregulation in cancer (Barbosa et al., 2021), suggesting the extensive effects of SD on physiological functions.

The genes enrichment analysis on 29 circadian rhythm-related genes closely related to SD found significantly enriched terms, including muscle structure development, cellular response to extracellular stimulus, selenium metabolism or selenoproteins, and circadian rhythm. This indicates that sleep has an impact on both muscle (Chennaoui et al., 2021) and metabolism (Gümüştekín et al., 2004) through circadian rhythm-related genes. These findings also provided a mechanism for SD on the inhibition of myocardial ejection. Encouragingly, seven genes, including Dicer1, Xbp1, Srebf1, Crem, Top1, Sfmbt1, and Naglu, are directly enriched in circadian rhythm. Although studies have reported the role of Dicer1 (Tinarelli et al., 2014), Xbp1 (Brown, Strus & Naidoo, 2017), Crem (Wimmer et al., 2021), and Naglu (Değerliyurt et al., 2021) in SD, their relationship with memory remains unclear, as well as their relationship with immunity. Srebf1 is a gene involved in lipid metabolism and has clinical significance in human blood (Lech et al., 2016). The inhibition of Top1, recently reported in severe acute respiratory syndrome coronavirus 2 (SARS-CoV-2), can play a large part in treating fatal inflammation and reducing mortality in mice and may provide an ideal therapy for inflammatory injury caused by SD (Ho et al., 2021). To our knowledge, the role of Sfmbt1 in sleep has not yet been explored.

SD interferes with memory function and physiological circadian genes, as evidenced by the declined memory function present in the water maze experiment. Seven circadian genes significantly changed in the brain with SD. Some genes also have potential value in different organs. Dicer1 and Xbp1 also have clinical significance in human blood. BDNF (Lee, Everitt & Thomas, 2004) and NPAS4 (Ramamoorthi et al., 2011) are two classic indicators of memory formation and consolidation, contributing to hippocampal synaptogenesis and plasticity. Consistent with our prediction, a correlation analysis showed that seven circadian genes, related to BDNF and NPAS4, were the essential components of memory function. Combined with previous studies, we have sufficient evidence to believe that these genes change memory function.

The changes in the immune environment were the dominant determinant of memory in the hippocampus. Among immune cells, the expression levels of M2 macrophages and naïve B cells were the highest in the hippocampus. M2 macrophages are part of immune cells that accelerate the progression of various tumors (Garrido et al., 2020), along with promoting the termination of neuroinflammation and improving cognitive function (Zhu et al., 2016). Although M2 macrophages play a decisive role in memory, they showed no changes in the hippocampus after SD. Interestingly, naïve B cells, follicular helper T cells, activated NK cells, and activated dendritic cells changed significantly after SD, indicating a potential link with SD. Increased naïve B cells in the blood after exercise promotes attention to learning test (Sîrbulescu et al., 2019) Similar to the changes seen after SD, the content of follicular helper T cells also decreased in patients with psoriasis (Shin et al., 2016). Patients with Parkinson’s Disease also showed activated NK cells (Zhou et al., 2020). In addition, the reduction of dendritic cells in the peripheral blood of patients with Alzheimer’s disease has been shown to exacerbate clinical symptoms (Ciaramella et al., 2016). Our results revealed that four immune cells changed in the hippocampus, which further elucidates the mechanism of SD in regulating the immune environment on memory.

Finally, we explored the effects of circadian genes on the immune environment. Although studies have conclusively shown that circadian genes influence the immune environment (Boivin et al., 2003), intensive studies of SD still lag behind. We found multiple relationships between circadian genes and immune cells, some of which have appeared in other diseases. The inactivation of Dicer1 leads to NK cell differentiation dysfunction (Degouve et al., 2016). Srebf1 activates NK cells in melanoma (Fu et al., 2020). SD inhibits both Srebf1 and NK cells, according to their regulatory relationship. The other regulatory relationships we found in this study have not been reported in SD. These regulatory relationships indicate the need for further research and deserve further confirmation.

This study creatively investigated the effects of circadian genes and hippocampal immune environment on memory function after SD and elucidated the relationship between circadian genes and immune cells. This study does still have some limitations. First, the model condition of SD from the two datasets we used was not the same. For example, the duration of SD was different, which may increase the difference between the two sequencing results. Second, the samples used of the two datasets were from the brain and the hippocampus. The brain contains many organized regions, which increases the difficulty and possible inaccuracy of the analysis. Therefore, combined with the relationship between memory and the hippocampus, we only detected the immune infiltration of the hippocampus. Third, the expression profiles only showed Dicer1 and Xbp1 in human blood because of the different sequencing data platforms, but the role of other circadian clock genes in humans is still worth further study. Fourth, although we demonstrated the significance of naïve B cells in GSE34424, we are still lacking effective experimental verification. Further experimental studies are required to illuminate the relationship between circadian genes and immune cells.

Conclusions

In summary, a bioinformatics analysis of circadian genes and the hippocampal immune environment reveals that their relationship with memory may play an essential role in SD memory impairment, providing potential targets for SD treatment and profound insights into its genetic mechanisms.

Supplemental Information

Supplemental Information 1 Differentially expressed genes in two datasets

Click here for additional data file.

Supplemental Information 2 The results of GSEA enrichment

Click here for additional data file.

Supplemental Information 3 Gene list of circadian rhythm-related genes obtained from GeneCards

Click here for additional data file.

Supplemental Information 4 ROC curve of genes in GSE33302

Click here for additional data file.

Supplemental Information 5 Expression profiles of GSE33302

Click here for additional data file.

Supplemental Information 6 Expression profiles of GSE9442

Click here for additional data file.

Supplemental Information 7 Expression of naive B cells in GSE34424

Click here for additional data file.

Supplemental Information 8 Raw data of the water maze experiment

Click here for additional data file.

Supplemental Information 9 Full (21-point) ARRIVE 2.0 checklist

Click here for additional data file.

We thank the friends of the Han group in room 445 from the Pharmaceutics of Zhejiang University for their encouragement. Also, we would like to thank PeerJ staff for English language editing.

Additional Information and Declarations

Competing Interests

Author Contributions

Animal Ethics

Data Availability

The authors declare there are no competing interests.

Peng Ke, Chengjie Zheng, Xiaodan Wu and Kai Zeng conceived and designed the experiments, performed the experiments, analyzed the data, prepared figures and/or tables, authored or reviewed drafts of the paper, and approved the final draft.

Feng Liu, LinJie Wu and Yijie Tang performed the experiments, analyzed the data, authored or reviewed drafts of the paper, and approved the final draft.

Yanqin Wu, Dongdong Lv, Huangli Chen and Lin Qian analyzed the data, authored or reviewed drafts of the paper, and approved the final draft.

The following information was supplied relating to ethical approvals (i.e., approving body and any reference numbers):

Committee of experimental animals of Fujian Medical University provided full approval for this research (FJMU IACUC 2021-0458)

The following information was supplied regarding data availability:

The raw data are available in the Supplementary Files.

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
