# Peer review of "Relationship between circadian genes and memory impairment caused by sleep deprivation"

_PeerJ, doi:10.7717/peerj.13165_

## Round 0.1 · original submission · Major Revisions

Please address the concerns of both reviewers and amend the manuscript accordingly.

Reviewer 1 ·

Basic reporting

1. There are numerous grammatical errors throughout the manuscript that should be addressed.

For example, in lines 94-97: “An innate circadian clock, a kind of 24 h cell-autonomous period in human-driven by transcription-translation feedback loops of circadian clock genes (Ikegami et al., 2019), controls daily activities and maintains the stable rhythm of organ function in many physiological and pathological processes (Patke et al., 2017).”

Additionally in 97-99: "In recent years, circadian rhythm has been concerned by many neurological disorders, including schizophrenia, bipolar disorder, depression, and autism (Johansson et al., 2016)."

Errors like these are distracting and should be corrected prior to resubmission.

2. Additionally, the section in the intro concerning circadian rhythm biology is overly basic and should include more information about the clock genes assessed in the study.

Experimental design

1. The experimental design is reasonable in this reviewer's opinion, although again clarity is obscured at times due to grammatical errors and incomplete explanations.

2. It is unclear what SD protocols were used and how the control mice were treated in the data not collected by this group. Although a reference is provided for these experiments, a short explanation of how these studies were conducted would be useful.

Validity of the findings

1. 295-296 The authors state, “The important MCODE module in the PPI network, circadian rhythm genes marked in pink, showed seven circadian clock genes (Fig. 3D-E). None of the genes listed function as part of the circadian clock, although these may be output genes from the central oscillator. Based on this and the intro section about circadian rhythm biology, it appears as though the authors lack a solid understanding about how the circadian clock functions and it is recommended that they consult a clock expert as part of the revision process.

2. It is unclear at times which data set the authors are referring to at times. The experiments should be better introduced in the results section and perhaps a brief explanation as to why they were performed.

3. 318 - 323 “Therefore, we detected the expression of genes in the heart, liver, and lung, observing that some genes alternated in various organs (Fig. 5F). Seven circadian genes significantly and consistently correlated with two representative indicators of memory function, BDNF, and NPAS4. These results indicated that SD leads to memory impairment and changes of seven circadian clock genes, of which the latter were closely related to memory function.” Here, the authors assert that changes in clock-regulated (presumably) genes are associated with changes in two memory genes. However, it is unclear how this correlation can be made given that the gene changes did not occur in the brain.

4. The authors use changes in expression data to detect changes in immune cell infiltration into the hippocampus using the algorithm CIBERSORT. . Although potentially interesting, this data is highly correlative and should be confirmed empirically.

Additional comments

The authors seek to further establish the relationship between SD, circadian functionality and the immune system. This is a highly relevant field of research in the sleep and circadian community, which makes this article potentially impactful. However, numerous grammatical errors and a lack of understanding of circadian rhythms, and overstatement of the authors’ findings detract from the paper and should be addressed prior to submission.

Reviewer 2 ·

Basic reporting

In this study, the authors aim to investigate the relationship between circadian clock genes and the alteration of memory function after sleep deprivation (SD). The conclusion is mostly supported by the results shown. However, the authors should address the comments below and re-write the conclusion section of the manuscript (details below).

Authors should use clear and unambiguous English language throughout the manuscript. One such example is lines 84-86 in the introduction. What is emotional processing speed? "Numerous studies have shown that SD has deleterious effects on cognitive function through neurological damage and neurodevelopmental disorders (Sabia et al., 2021), including attention, working memory, emotional processing speed, and reaction (Turner et al., 2007)"

Experimental design

Line 141-142 "In addition, validation datasets (the entire datasets of GSE9442 and other related datasets) and animal experiments enhanced the reliability of the analysis results" How the entire dataset helped in enhanced reliability of the results is not at all clear. The authors should explain this statement and provide details in the method section as appropriate.

Paragraph 154-163 is poorly written and how the GSEA analysis was conducted is unclear. Please rewrite and emphasize how the analysis was done using the web-based toolkit (providing appropriate references)

Validity of the findings

Lines 261-269 Authors should describe in detail the results they obtained, rather than just referring to the figures.

I want the authors to comment on this whether we can reverse the course of memory impairment and other effects caused by SD by introducing a healthy sleep regime? Did they perform any experiments in which they tried to induce a healthy sleep regime on rats (previously suffering from memory impairment) and check their cognitive abilities over time? Is memory impairment caused by SD reversible?

---

## Round 0.2 · Major Revisions

As indicated by both reviewers, your manuscript require significant editorial work as it contains multiple linguistic issues and grammatical errors. Therefore, it is strongly advised to look for the professional editorial help. For example, PeerJ can provide language editing services for fees - please contact at [email protected] for pricing (be sure to provide your manuscript number and title).

Reviewer 1 ·

Basic reporting

see comments below

Experimental design

see comments below

Validity of the findings

see comments below4

Additional comments

Although the authors made significant improvements, this article still needs quite a bit of work. Specifically, numerous grammatical issues remain that make the article difficult to navigate. I highly recommend this group consults an outside group for help with this.

Reviewer 2 ·

Basic reporting

The authors have revised the manuscript in regards to the grammatical errors and English language. With they seeking help from language editing services from PeerJ will help improve this manuscript even further.

Experimental design

More details on the methodology are now added.

Validity of the findings

Satisfactory clarifications are provided to the concerns raised in the previous version.

---

## Round 0.3 · accepted · Accept

Thank you for fixing the remaining issues. Since all the concerns are addressed now and since the manuscript was carefully edited, the amended version is acceptable now.